# The use and acceptability of preprints in health and social care settings: A scoping review

**Amanda Jane Blatch-Jones**⬤*, **Alejandra Recio Saucedo, Beth Giddins**⬤

National Institute for Health and Care Research (NIHR) Coordinating Centre, School of Healthcare Enterprise and Innovation, University of Southampton, Southampton, Hampshire, United Kingdom

* A.J.Blatch-Jones@southampton.ac.uk

## Abstract

### Background

Preprints are open and accessible scientific manuscript or report that is shared publicly, through a preprint server, before being submitted to a journal. The value and importance of preprints has grown since its contribution during the public health emergency of the COVID-19 pandemic. Funders and publishers are establishing their position on the use of preprints, in grant applications and publishing models. However, the evidence supporting the use and acceptability of preprints varies across funders, publishers, and researchers. The scoping review explored the current evidence on the use and acceptability of preprints in health and social care settings by publishers, funders, and the research community throughout the research lifecycle.

### Methods

A scoping review was undertaken with no study or language limits. The search strategy was limited to the last five years (2017–2022) to capture changes influenced by COVID-19 (e.g., accelerated use and role of preprints in research). The review included international literature, including grey literature, and two databases were searched: *Scopus and Web of Science (24 August 2022)*.

### Results

379 titles and abstracts and 193 full text articles were assessed for eligibility. Ninety-eight articles met eligibility criteria and were included for full extraction. For barriers and challenges, 26 statements were grouped under four main themes (e.g., volume/growth of publications, quality assurance/trustworthiness, risks associated to credibility, and validation). For benefits and value, 34 statements were grouped under six themes (e.g., openness/transparency, increased visibility/credibility, open review process, open research, democratic process/systems, increased productivity/opportunities).

### Conclusions

Preprints provide opportunities for rapid dissemination but there is a need for clear policies and guidance from journals, publishers, and funders. Cautionary measures are needed to

**Data Availability Statement:** All relevant data are within the paper and its Supporting Information files. These files are also openly accessible on the preprints server.

**Funding:** This research was funded by the National Institute for Health and Care Research Coordinating Centre (NIHRCC), based at the University of Southampton, through its Research-on-Research Programme. The views and opinions expressed in the discussion are those of the authors and do not necessarily reflect those of the Department of Health and Social Care. The funders had no role in the study design, data collection and analysis, decision to publish, or preparation of the manuscript.

**Competing interests:** The authors have declared that no competing interests exist.

**Abbreviations:** APC, Article Processing Charges; ATOPP, Attitudes Towards Open data sharing, Preprinting and Peer-review; CIHR, Canadian Institutes of Health Research; COI, Conflict of Interest; CONSORT, Consolidated Standards of Reporting Trials; COVID-19, Coronavirus Disease 2019; COS, Center for Open Science; CRediT, Contributor Role Taxonomy; CV, Curriculum Vitae; DOI, Digital Object Identifier; DORA, Declaration on Research Assessment; DURC, Dual Use Research of Concern; ECR, Early Career Researchers; HEI, Higher Education Institution; JBI, Joanna Briggs Institute; LMICs, Low-and-Middle Income Countries; NIH, National Institutes of Health; NIHR, National Institute for Health and Care Research; OA, Open Access; ORCID, Open Researcher and Contributor IDentifier; PRISMA-ScR, Preferred Reporting Items for Systematic Reviews and Meta-analyses extension for Scoping Reviews; UK, United Kingdom; UKRI, UK Research and Innovation.

maintain the quality and value of preprints, paying particular attention to how findings are translated to the public. More research is needed to address some of the uncertainties addressed in this review.

## Background

The publication of research is slow, and accessibility of research outputs is often delayed due to manuscripts going through a lengthy process involving peer-review, revisions and getting published online [1]. Once published, the outputs of the research are not always widely accessible due to subscription charges, which ultimately places restrictions on who can access published manuscripts, especially those in resource-constrained countries or institutions, and the public. These barriers cause risk to the value of the research and its contribution to future scientific research and discovery being realised [2–5]. Preprints can offer a solution, and for some academic disciplines preprints have been part of the publication pathway for more than 20 years, providing instant open access to research via preprint servers (e.g., arXiv, bioRxiv, chemRxiv, and EarthArxiv) [1].

The first preprint server known as arXiv came into practice in the physics community in 1991 [6]. Posting a manuscript in a preprint server has its advantages particularly when it can take a long time to publish research findings in a peer review publication. Preprints allow researchers to post their research findings earlier and to allow for wider research community review that would otherwise only be afforded to a minimum number of peer reviewers (chosen by journal editors as part of the publication process) [2].

However, the acceptability and role of preprints can vary not only by discipline (including whether the research is biomedical, preclinical or implementation research) but also among researchers, publishers, and funding organisations.

### What are preprints?

Preprints are open and accessible scientific manuscripts that are shared publicly, through a preprint server, before being submitted to a journal (or at the time of submission to a journal, as part of the submission process of the journal). Sharing a manuscript or report publicly prior to submission enables full access and, increases equity in the publication process (providing the ability to share research without paying access charges). However, it is important to note that some proponents of preprinting may only publish their work as a preprint and not as an article in a peer reviewed journal.

### Use and acceptability of preprints

The role of preprints during the COVID-19 pandemic highlighted their importance in a public health emergency, and now with their continued use, it is important to understand how preprints contribute to the open research agenda [7–10]. However, the acceptability and use of preprints by some in the research community, can lead to hesitancy or resistance from researchers to submit their research to a preprint server [11, 12]. Concerns around the integrity and validity of submitting a manuscript to a preprint server, is also a challenge for articles published in a peer reviewed journal [13, 14]. Peer reviewed journal articles can also be retracted due to unreliable findings or errors in the published article. Funding organisations have different policy statements about how and where preprints are accepted as part of the application process. For example, whether preprints can be referenced as part of a grant

application; included in an applicant's curriculum vitae (CV); or form part of the dissemination process and publication of findings.

Understanding the position of funding organisations on preprints has important implications for how the research community is willing to engage in, and accept the added value and benefit of, publishing their research in a timely fashion, and, contribute to advancing scientific knowledge [12, 15, 16]. Some funding organisations, such as Wellcome and UK Research and Innovation (UKRI) have already initiated policy guidance on their position of preprints in grant applications and some journal publishers (e.g., Springer Nature, Wiley and SAGE) are beginning to implement statements around their publishing models.

The purpose of this systematic scoping review was to explore the position of publishers, funding organisations and the research community in terms of the use and acceptability of preprints in health and social care settings at the grant application stage and publication of findings (e.g., the use of preprints in grant applications or as part of the publishing model). To our knowledge, a review of this topic or at this scale has not been previously undertaken. The review was conducted to address the following question: *What does the evidence say about the role and acceptance of preprints (challenges, benefits, value, hesitancy, impact) throughout the research lifecycle (e.g., grant applications and publication of findings)*?

## Methods

A previous non-systematic search identified several articles examining the role of preprints and how preprints contribute to the promotion of the open research agenda. This preliminary search was used to inform the design and conduct of this systematic scoping review (a proforma was written prior to conducting the review to ensure the rationale, objectives, design, and methodology were followed).

Due to the complexity, uncertainty, and nature of the available research (in terms of source, type, and audience) a scoping review methodology was undertaken. Scoping reviews are relevant to addressing research questions on priorities for research, clarification on concepts and definitions, and, providing research frameworks or background information in preparation for a systematic review. Scoping reviews typically identify evidence gaps or scope the body of literature rather than seeking to describe experiences or current practice [17]. Scoping reviews look to address 'what has been done previously?' and 'what does the literature say?' compared to systematic reviews that ask the question 'does this intervention work for this particular group?'. Scoping review methodology does not judge the quality of the evidence but rather intends to map the evidence. Although scoping reviews present an overview of a broad research question or topic from a diverse body of evidence, they are conducted using methodological frameworks and guidance [18, 19]. The Joanna Briggs Institute (JBI) guidance was used to support the development, analysis and write up of the scoping review [20–22].

### Eligibility criteria

**Context.**   The context included UK and international settings within the academic environment.

**Participants.**   Publishers, research funding organisations and Higher Education Institutions (HEI's) use of preprints were included (e.g., academia or research management focused). The role of preprints from a researcher's perspective, placing emphasis on open science and open research, was also considered.

**Inclusion criteria.**   Evidence from research disciplines that focused on health and social care and all phases of research, from the point of testing a new treatment or intervention through to implementation and service delivery research were included. This is not an

exhaustive list but included health and social care disciplines such as sociology, biology, health economics, social work, nutrition, chemistry, pharmacology, and psychology.

**Exclusion criteria.** Evidence outside of the health and social care discipline, (e.g., engineering, physics, agriculture, arts, music) including industry and private businesses with no academic focus. The authors limited the review to only this discipline due to the variation of perspectives across other disciplines (including acceptability and how preprints are used to inform future scientific discovery) and to explore what challenges were specific to health and social care that may not be relevant to other fields of research. Non-English articles were excluded if no translation was available for the full article.

## Types of sources

The scoping review considered all types of study designs for inclusion (e.g., randomised controlled trials, non-randomised controlled trials, before and after studies and interrupted time-series studies, analytical observational studies including prospective and retrospective cohort studies, case-control studies, analytical cross-sectional studies, descriptive observational study designs including case series, individual case reports and descriptive cross-sectional studies).

Qualitative studies that focused on, but not limited to, designs such as phenomenology, grounded theory, ethnography, qualitative description, action research and feminist research were also considered. In addition, systematic reviews, text, and opinion articles that met the inclusion criteria were also considered.

The review included several types of published material such as academic outputs through journal articles, commentaries, editorials, and opinion letters. Grey literature of policies, guidance and reports from funding organisations and publishers were also included. It was important to capture all types of literature due to the nature of the research question and to determine whether there were any gaps in the evidence (is the evidence anecdotal or were there research studies on the role and acceptance of preprints?).

## Search strategy

The search strategy aimed to locate both peer reviewed and non-peer reviewed studies (through the inclusion of conference abstracts). An initial limited search of PubMed, Mendeley and Google Scholar was undertaken to identify articles on the topic. The text contained in the titles and abstracts of relevant articles and reports, and the index terms used to describe the articles were used to develop a full search strategy. The search strategy, including all identified keywords and index terms, was adapted for each database **(See S1 Appendix: Search terms and keywords; S1 Table: Scopus and Web of Science searches)**. The reference lists of included articles were also screened to identify additional literature material.

There were no study or language limits applied during the information retrieval process. However, the search strategy was limited to the last five years (2017–2022), which enabled us to determine whether the COVID-19 pandemic accelerated the use and role of preprints in health and social care research. Two databases were searched: Scopus and Web of Science on 24 August 2022.

## Data extraction and evidence selection

Following the search, all identified articles were collated and uploaded into EndNote version 20 *(Clarivate Analytics*, *PA*, *USA)* and duplicates removed. A pilot test of titles and abstracts were then screened for assessment against the inclusion criteria. Relevant articles were retrieved in full for screening. Ten articles were screened by all reviewers (ABJ, ARS, BG) to ensure continuity and agreement across the team. All articles were reviewed independently by

two reviewers (ABJ reviewed all articles and BG and ARS reviewed 50% each) and notes were included in a separate Endnote library. Where the independent reviewer was unsure of the inclusion of the article, the record was annotated for discussion with the team and decisions for inclusion or exclusion were made by consensus. Disagreements between the reviewers at each stage of the selection process were also resolved through discussion.

After the final screening, the list of included articles was exported to an Excel spreadsheet where labelling of articles matched the evidence to:

- the two processes of interest (funding application and/or publication of findings),

- specific areas of interest (challenges, barriers, benefits, solutions, recommendations).

The results of the search and the study inclusion process are reported in full in a Preferred Reporting Items for Systematic Reviews and Meta-analyses extension for scoping review (PRISMA-ScR) flow diagram (see **Fig 1**).

Data were extracted from the articles included in the scoping review by using a data extraction tool developed by the reviewers, based on the research question. The tool included specific details about the participants, concept, context, study methods and key findings. The extraction included direct export from the included articles into the spreadsheet to avoid having to return to the original source (and prevent subjective or evaluative accounts of the included articles). Extraction from each full text article was divided between the three reviewers and

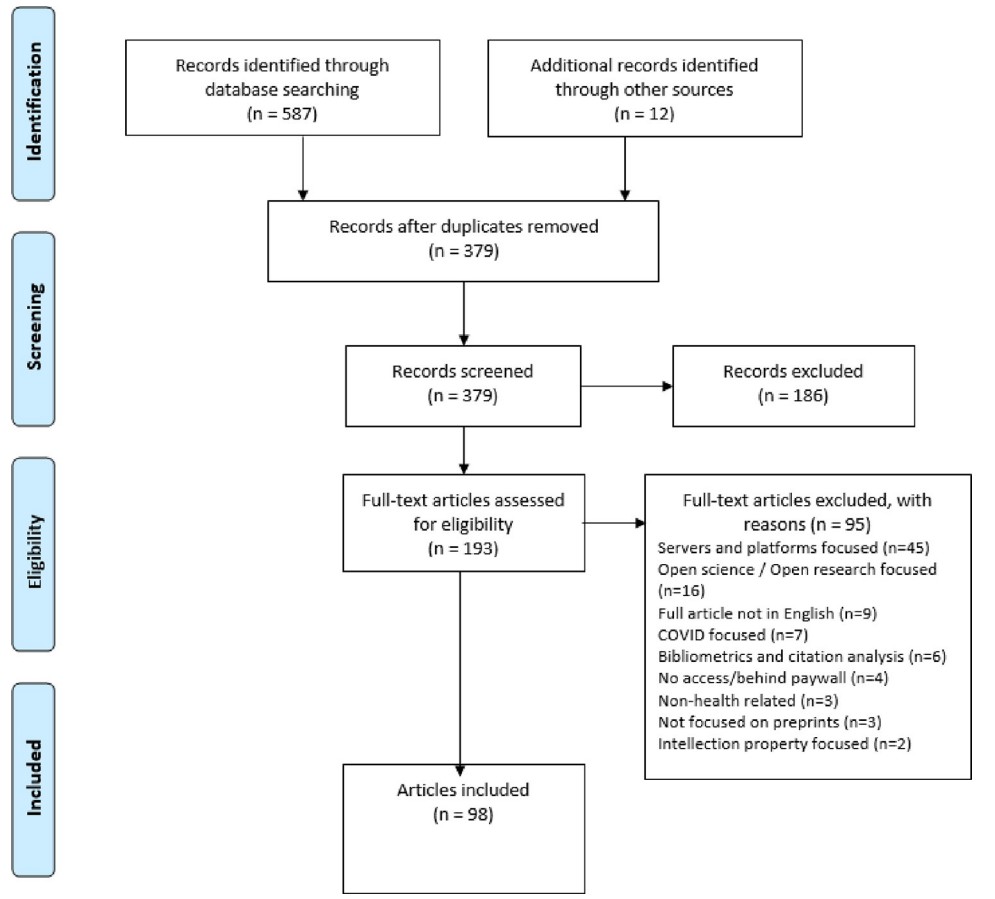

**Fig 1. Flow diagram of the included articles.**

ABJ reviewed and verified all extraction. Due to using a scoping review methodology, no risk of bias or assessment on quality was conducted. All the evidence was mapped and categorised to specific terms, which were discussed and agreed between team members at various stages throughout data extraction.

Funding organisations (e.g., Wellcome, UKRI, National Institute for Health and Care Research (NIHR), National Institutes of Health (NIH), Canadian Institutes of Health Research (CIHR), European Research Council)), including those organisations referenced in key articles, were explored as part of the grey literature search using Google. A search of publishers (e.g., Springer, Wiley, SAGE, PlosOne, F1000) were also conducted for additional information to understand the policy, position, and acceptability of preprints in their publication process. The policy, guidance and publications webpages were visited, and any preprint server details were recorded. The purposive sampling of funding organisations and publishers were drawn from discussions with NIHR Coordinating Centre staff to better understand the position of other funding organisations and publishers. A separate excel spreadsheet was used to capture the search results, specifically focusing on funding organisations and publishers preprint policies, guidance, and statements.

## Results

A total of 587 articles were retrieved from the two databases. After deduplication (208 articles), 379 titles and abstracts and 193 full text articles were assessed for eligibility (12 articles were retrieved from references of those included articles). Of these 193, 98 articles met the eligibility criteria and were subsequently included in the review for full extraction. **Fig 1** provides a full account of the records of identification flow diagram, including the reasons for the excluded articles. **(See S2 Table: Full details of the 98 included articles).**

### Characteristics of the included studies

All articles were published after 2017. Most of the articles were published in a journal with a majority being perspective, editorial or commentaries (n = 69). There were 24 original research articles that consisted of surveys with researchers, clinicians, and editors; analyses of preprint servers and policies; analyses of funding organisations and publishers preprint policies; and comparisons of preprints to peer reviewed publications. Characteristics of the 98 included articles are presented in **Table 1.**

As expected, much of the evidence focused on the publication stage of research with minimal evidence on the use of preprints at the grant application stage, apart from providing statements that some funding organisations accept preprints as part of a grant application. More than half of the evidence was published from 2020 onwards (n = 63) and most of the articles (based on first authors country) were based in Europe (n = 36) or Americas (n = 35).

### Summarising the evidence

To summarise the evidence found in the database searches, key statements were derived from the articles and labelled under generic categories. **Table 2** presents the statements, along with frequencies and source of evidence on the challenges and barriers associated to preprints. **Table 3** presents the statements, along with frequencies and source of evidence on the benefits and value associated with the use of preprints.

**Fig 2** provides an overview of the benefit and value, and challenges and barriers statements associated to the use, acceptability, and implementation of preprints.

All the statements associated with the use of preprints varied considerably depending on what perspective was taken (publishers, funding organisations or researchers) and when the

**Table 1. Summary of the characteristics of the included articles (n = 98).**

| Characteristics | N (%) |
| --- | --- |
| *Area of focus*: | |
| Publications only | 90 (91.8) |
| Grants only | 0 |
| Publications and Grants | 8 (8.2) |
| *Year of publication*: | |
| 2017 | 11 (11.2) |
| 2018 | 10 (10.2) |
| 2019 | 14 (14.3) |
| 2020 | 19 (19.4) |
| 2021 | 30 (30.6) |
| 2022* | 14 (14.3) |
| *Country by region*:** | |
| Africa | 1 (1.0) |
| Asia | 12 (12.3) |
| Europe | 36 (36.7) |
| Americas | 35 (35.7) |
| Australia | 5 (5.1) |
| Unknown | 9 (9.2) |
| *Article type*: | |
| Journal–Original research | 24 (24.5) |
| Journal–Review | 4 (4.1) |
| Journal–Perspective*** | 69 (70.4) |
| Conference proceeding | 1 (1.0) |

*Jan-Aug inclusive, search was conducted on 24[th] August 2022

** Americas (Brazil, Canada, Cuba, Mexico, USA)

***Includes, editorial, commentaries, news features, correspondence, and perspective articles in journals.

article was published. There was a steady increase in the number of publications on preprints since 2020, which could be associated to the COVID-19 pandemic. However, any association between the effect of COVID-19 and the volume of preprints requires further research, which was out of remit for this scoping review (scoping methodology does not assess risk of bias or quality).

The remaining sections provide a summary of the evidence based on the research question: What does the evidence say about the role and acceptance of preprints throughout the research lifecycle, specifically in grant applications and publication of results?.

## Barriers and challenges associated with preprints

Although preprints have been established in some disciplines for more than 20 years, the COVID-19 pandemic has revealed the rapid progress of preprints, both for their value and their shortcomings [8, 10, 23–39]. Preprints are considered to lack scientific integrity, credibility and quality, and there is still reservation among editors, publishers, journals and researchers about the role of preprints and their contribution to science [4, 6, 35, 40–43].

The evidence provided several process issues related to why preprints lack quality, screening, or quality checks during the submission of a manuscript to a preprint server [33, 44, 45]. It was reported that only three preprint servers (Research Square, bioRxiv and medRxiv) check if the content contains unfounded medical claims [33, 44]. Several articles suggested that

**Table 2. Challenges and barriers associated to preprint publications, ranked by theme and numerical frequency.**

| Key themes | No | For whom? | Statements | No. refs |
|---|---|---|---|---|
| **Volume and growth of publications (replication crisis)** | 1 | Researchers, Funders, Publishers | Increasing level of literature risks becoming a dumping ground for unreliable results, making it difficult to separate signal from noise (with speed comes errors). Scientific progress will slow and be put at risk as a result [23, 32, 37, 46, 47, 55]. | 6 |
| | 2 | Researchers | Increases the volume of literature, resulting in increased time to review and discover quality evidence (mishmash of papers of various quality; information overload; information overlap) [6, 12, 29, 30, 37, 56]. | 6 |
| | 3 | Researchers and Publishers | A preprint platform may require authors to post the manuscript under a particular license that could conflict with the license / copyright transfer agreements required by a journal [2, 54, 57, 58]. | 4 |
| | 4 | Researchers and Publishers | Not all preprints end up in a peer reviewed journal (posting of a preprint could forestall a publication in a peer reviewed journal) [6, 53, 55, 57]. | 4 |
| | 5 | Researchers | Increased "junk science" in the public domain (including multiple versions) could affect benchmarking indices / metrics of scholarly achievements (e.g., h-index, altimetric) causing confusion for readers and potentially harming research teams [23, 30, 59, 60]. | 4 |
| | 6 | Publishers | Preprints can be seen as a duplicate publication, namely "dual-use research of concern" (DURC) [30, 46, 48]. | 3 |
| | 7 | Publishers | Preprints can be seen to be a threat to the multi-billion-dollar publishing oligopoly [13, 15]. | 2 |
| **Quality assurance and trustworthiness (including misuse)** | 8 | Researchers, Funders, Publishers | No scrutiny of preprints means no quality assurance of the content of articles (lack of quality control, threatening the quality) increase potential for misuse, misconduct, and fraud [6, 14, 24, 26, 33, 35, 37, 46, 61–63]. | 11 |
| | 9 | Researchers, Funders, Publishers | Premature media coverage of preprints and public sharing of information prior to any scrutiny or peer review (public might not understand the role and limitations of preprints in knowledge dissemination) [4, 7, 25–27, 30, 32, 49, 51]. | 9 |
| | 10 | Researchers, Funders, Publishers | Preprints should not be used to guide clinical practice or health-related behavior and should not be reported in news media as established information [7, 8, 26, 50, 51, 64]. | 6 |
| | 11 | Researchers, Funders, Publishers | Preprints can have a negative impact on credibility and public perception towards research (what if a preprint with a potential impact on public health is interpreted by some as established evidence?); preliminary findings being made available to the public in a way that appear credible [2, 4, 6, 30, 46, 64]. | 6 |
| | 12 | Publishers | Needs to be clear, consistent and transparent policies on preprints (including policies in preprint servers), otherwise there will be distrust and the growth, acceptance of preprints will be minimal (requires integrity) and unclear preprint policies could hinder uptake of open research practices (disparity between disciplines results in unclear policies between publishers) [35, 40, 41, 42, 43, 57]. | 6 |
| | 13 | Researchers, Funders, Publishers | Security and screening of preprints is lacking during the scientific process leading to the potential of disseminating misleading information (at pre and post study stages) mitigation of risk is required due to lack of validation (authors could inflate key findings as no checks) [6, 33, 36, 65, 66]. | 5 |
| **Risks associated to credibility and reputation (including unscholarly behaviour)** | 14 | Researchers and Publishers | Preprints provide limited protection and allows for the publication of poor-quality articles on preprint servers (including plagiarism) [4, 6, 15, 36, 39, 44, 47]. | 7 |
| | 15 | Researchers | Intellectual 'scooping' is seen as a risk for researchers [2, 12, 15, 47, 52–54]. | 7 |
| | 16 | Researchers and Publishers | Peer review is a vital step in preserving the scrutiny and integrity of scientific information and is valued by authors (risk of reproducibility leading to significant differences between preprints and published versions) [25, 50, 51, 63, 67, 68]. | 6 |
| | 17 | Researchers and Publishers | Peer-review and journal reputational cues are lacking from preprints. They are seen as important pre-conditions for researchers to engage with scholarly work (including significance of the work) but lacks credibility [9, 67]. | 2 |
| | 18 | Publishers | Funding for preprint servers is from non-profit agencies and concerns have been raised regarding sustainability and archiving costs [41]. | 1 |
| | 19 | Researchers, Funders, Publishers | Difficult to disseminate or promote preprints to broader research community without jeopardising anonymity (underrepresented racial, socioeconomic, gender, or sexual identity groups may experience negative bias from a lack of anonymity) [59]. | 1 |

*(Continued)*

**Table 2.** (Continued)

| Key themes | No | For whom? | Statements | No. refs |
|---|---|---|---|---|
| **Scientific validation / quality** | 20 | Researchers, Funders, Publishers | Selective use of preprint servers and exercising caution for clinical investigators when the focus is on a study involving drugs, vaccines, or medical devices and results may directly affect treatment of patients (risk on human health) and public health and safety [8, 26, 32, 33, 35, 45, 46, 48–50]. | 10 |
| | 21 | Researchers, Funders, Publishers | Many are not aware of corrections by retractions or errata leading to misuse of research, fraud, and misconduct (risk during pandemic periods) [6, 14, 29, 32, 35, 39, 46, 55]. | 8 |
| | 22 | Researchers, Funders, Publishers | Preprints could influence policy and guideline decisions with practical ramifications / implications, including ethical dilemmas for medical professionals and government (research integrity) [6, 23, 24, 27, 45, 54, 67]. | 7 |
| | 23 | Researchers and Publishers | Preprints are an initial step along the scientific dissemination and publication pathway—cautionary warnings are required (indicating that this is preliminary research that has not been peer reviewed, an interim research product) [25, 26, 33, 64]. | 4 |
| | 24 | Researchers, Funders, Publishers | Readers assume the version in the preprint server is the final version (including causing confusion as to the version of record) and there could be factually incorrect information [6, 15, 59]. | 3 |
| | 25 | Researchers and Publishers | Lack of clarity from journals and publishers can prevent researchers from using preprints as a source to disseminate their research findings (journals reject studies already posted as a preprint; low visibility) [12, 53, 65]. | 3 |
| | 26 | Researchers and Publishers | Not all disciplines or fields require rapid sharing of results (pace should not be prioritised over quality and scrutiny) [4, 46]. | 2 |

**Table 3. Benefits and value associated with the use of preprint publications, ranked by theme and numerical frequency.**

| Key themes | No | For whom? | Statements | No. Refs |
|---|---|---|---|---|
| **Openness and Transparency** | 1 | Publishers and Researchers | When results are needed quickly, preprints offer a solution (including greater visibility on search engines) for faster dissemination than traditional publication routes (aggregated time saving could make scientific discovery 5 times faster over 10 years) Preprints accelerate the dissemination of research and innovation [1–4, 6–8, 23, 26, 28, 29, 33, 41, 43, 44, 46, 55, 58, 59, 69–74, 76, 78, 81–85]. | 33 |
| | 2 | Publishers, Funders and Researchers | Preprint servers can help mitigate positive-outcome bias and increase transparency and data sharing opportunities, including reproducibility (including generating new research questions, expanding results and usefulness) [5, 28, 29, 37, 46, 50, 55, 69, 86–88]. | 11 |
| | 3 | Publishers | Open and transparent history of versions of an article; provides access to previous versions (being free to read) allowing for different versions to be accessible (including datasets and protocols) [6, 26, 41, 50, 70, 78–81]. | 9 |
| | 4 | Publishers | Preprints allow for the promotion of replications, confirmatory, contradictory, or negative findings, which generally tend to be marginalised by traditional journals (including controversial results) including alternative platforms to present data [5, 15, 50, 58, 59, 69, 70]. | 7 |
| **Increased visibility / credibility** | 5 | Funders and Researchers | Preprints can further inform grant review and academic advancement (attractive for researchers) [2, 23, 41, 70, 75, 76, 78, 89, 90]. | 9 |
| | 6 | Researchers | Oversight earlier in research process, therefore greater visibility earlier on (and rapid reuse of research including data) offering new opportunities for an author's work to be used/cited and therefore demonstrating the impact of the research earlier on [1, 41, 58, 66, 70, 73, 91]. | 7 |
| | 7 | Researchers | Encourages research behaviour for data sharing, code sharing, pre-registration early on at the preprint stage, signaling increased credibility of preprints (including scientific validation) [9, 46, 55]. | 3 |
| | 8 | Researchers | Claim recognition over results immediately (in public domain and on the web) and to justify financial funds, for those scientific areas of high competition for development and of limited funding [4, 41]. | 2 |
| | 9 | Funders and Researchers | Allows funders to observe the progress of a project in real time, allowing a more realistic opportunity to apply for, and obtain funding or promotions [15, 43]. | 2 |

(Continued)

**Table 3.** (Continued)

| Key themes | No | For whom? | Statements | No. Refs |
|---|---|---|---|---|
| **Independent and open review process** | 10 | Publishers and Researchers | Preprint repositories allow for feedback from a wider population (including the public) continuing to complement traditional journal publishing, adding speed, openness and faster feedback for researchers [3, 4, 6, 41, 43, 46, 53–55, 58, 65, 76, 78, 81, 91, 92]. | 16 |
| | 11 | Publishers and Researchers | Preprints allow authors to widen their opportunities for receiving comments on their work by other researchers (and general public) with the goal of an improved final peer-reviewed publication / publication of record (including critical methodological review) [1, 26, 39, 43, 47, 50, 58, 59, 61, 64, 70, 71, 73, 76, 91, 93]. | 16 |
| | 12 | Publishers and Researchers | Freely accessible and freely reproducible preprints mean the value (e.g., quality) of scientific articles no longer needs peer review but is open to the evaluation and use of the whole scientific community [3, 12, 23, 29, 53, 59, 62, 69, 78, 81, 87, 90, 94]. | 12 |
| | 13 | Publishers and Researchers | Community response may not only improve manuscripts in development but also increase the efficiency and effectiveness of subsequent peer review by addressing inadequacies upstream (soliciting community feedback; verify quality of manuscripts, including plagiarism detection; scrutinises and critiques data, reducing the negative impact of problematic reports) [1, 3, 7, 29, 33, 50, 58, 69, 71, 73, 94]. | 11 |
| | 14 | Publishers | Preprints should be viewed as a working in progress, made open to the public for open feedback to improve the preprint (not to be confused with open peer review) [15, 23, 73]. | 3 |
| | 15 | Publishers and Researchers | Opportunities to accelerate discovery in ways not offered by traditional publishers to the contributing authors (early insight) but not to take the place of rigorous scientific evaluation [28, 31, 70]. | 3 |
| **Open research** | 16 | Publishers | Published under open access licenses enabling greater visibility and accessibility with no restriction or barriers (and no submission charges, lower cost, or at reduced cost, delays associated with peer review); virtually all preprint services offer 'green open access' [3, 4, 6, 9, 12, 44, 46, 63, 82, 86, 95]. | 11 |
| | 17 | Publishers and Researchers | Preprint model brings agility, free, unrestricted, and open access; assurance of originality, ensuring the priority of the discovery of research topic to the author, institution and research groups (allowing research to advance) [3, 69, 78, 86, 90]. | 5 |
| | 18 | Publishers and Researchers | Open science practices such as preregistration, open data, open materials, and preprints can help improve the rigor of research, and increase access to important materials, which can help disseminate quality work (and optimize research design and quality); researchers often careful to disclose their best work that reflects their scientific abilities and expertise, so work of low quality would not be expected [7, 23, 53, 59, 70]. | 5 |
| | 19 | Publishers, Funders and Researchers | Preprints offer a way of sharing important scholarly output that would otherwise disappear after much researcher time and funder expense (researchers are funded by and accountable to governments, the public, and funding organisations—research findings should remain accessible) [51, 70, 75]. | 3 |
| | 20 | Publishers | Avoid weeks/months for peer review and makes early results citable from the outset; a response to the replication crisis, an expression of the failure of traditional biomedical publishing [43, 76, 81]. | 3 |
| **Democratic process and systems** | 21 | Publishers and Researchers | The risk of scooping intellectual ideas such as methods from a preprint is unlikely because a preprint server offers time-sensitive evidence of an intellectual claim (un-editable timestamp within 24 hours of submission, providing a public record) [12, 15, 41, 53, 70, 76–78]. | 8 |
| | 22 | Publishers | Preprints are less biased as not subject to selectivity regarding which articles are published and preprints are 'unbranded' helping to refocus attention where it should be–not on the name of the journal where it is eventually published, but on its contents (addressing publication bias) [6, 39, 50, 56, 65, 73, 81]. | 7 |
| | 23 | Publishers and Researchers | Preprints can be cited, through the use of a Digital Object Identifier (DOI), so issue of responsible citation and use of preprints lies in the hands of authors and editors who choose to use and publish reference to preprint work (preprint permanently citable) [2, 3, 15, 23, 53, 76]. | 6 |
| | 24 | Publishers and Researchers | Feedback from a larger community compared to a typical peer-review that involves comments from two or three experts in the field reduces bias and increases diversity (including ECRs developing reviewer skills) [33, 53, 59, 73, 81]. | 5 |
| | 25 | Publishers and Researchers | Provide alternatives to underreporting, incomplete or misleading reporting by providing open access to the first finished compilation of the protocol–summary results–final dataset–preprint–paper series of prepublication events (enabling revision and improvements) [4, 29, 53, 69, 80]. | 5 |
| | 26 | Publishers | Greater democratic publishing process and preprints do not typically preclude publication (greater control over when work is made public); removing gatekeeper role that journals may play [6, 66, 70, 96]. | 4 |

(*Continued*)

**Table 3.** (Continued)

| Key themes | No | For whom? | Statements | No. Refs |
|---|---|---|---|---|
| **Increased productivity and opportunities** | 27 | Researchers | Preprints could facilitate interactions between researchers working on similar areas/projects, and help foster new collaborations and foster professional networking (including rapidity can help a researcher maintain enthusiasm and establish priority; ECRs connect with peers) [2, 3, 5, 12, 31, 32, 43, 50, 53, 58, 62, 64, 65, 68, 69, 76–78, 89]. | 19 |
| | 28 | Funders and Researchers | Providing early career researchers/scientists opportunities to showcase their work early to the public, offering positive effect on job opportunities or funding [5, 16, 43, 53, 62, 65, 75, 76, 78, 82, 89]. | 11 |
| | 29 | Funders and Researchers | Some funders allow inclusion of preprints in grant applications helping applicants and authors to provide evidence of research productivity (including communication and dissemination and for ECRs) [2, 23, 30, 41, 53, 76, 89, 90, 97]. | 9 |
| | 30 | Publishers and Researchers | Dissemination of healthcare research will facilitate better healthcare especially during a pandemic (offering priority to discoveries and ideas) [5, 7, 8, 23]. | 4 |
| | 31 | Publishers | Preprint platforms could provide opportunities to publishers/journal editors by scouting upcoming work and invite the submission of suitable manuscripts to their journal (special issues) [2, 12, 95]. | 3 |
| | 32 | Publishers | Preprint servers may offer a way to support living research documents and offer of results in real time [29, 51, 69]. | 3 |
| | 33 | Publishers and Researchers | Increase sharing and uptake of research from LMICs and use of preprint servers (no publishing cost) and open access (with waivers) publishing will increase accessibility of research; raise awareness in developing nations with limited institutional funds [23, 53, 84]. | 3 |
| | 34 | Funders and Researchers | Avoid spending time and resource on redundant research, therefore increasing efficiency, and reducing research waste [58, 69]. | 2 |

preprints represent incomplete scrutinised documents that have not been validated by peer review [14], there is higher risk of retractions [39], and not all preprints end up getting published in a peer reviewed publication [6, 32].

The wider implications of preprints not having the same or similar quality checks or assurances means that information is subject to interpretation and could be used by the public and social media inaccurately [32]. This may result in the public reading a preprint as 'established evidence' when it has not been through any peer review [2, 30, 46] and the media sensationalising the research before it is vetted as sound science [32, 47]. The lack of understanding of the role of preprints in the research process could equate to potential harm and have immediate impact on medical and clinical practice, including patients [8, 26, 32, 33, 35, 45, 46, 48–50]. Policy decisions may also be based on research where flaws have not yet been corrected as they typically would be during the peer review process [31, 51]. For researchers, the evidence suggested a high level of concern around the risks of intellectual 'scooping' [2, 12, 15, 47, 52–54] and for some disciplines or fields of research, rapid dissemination of results should not be

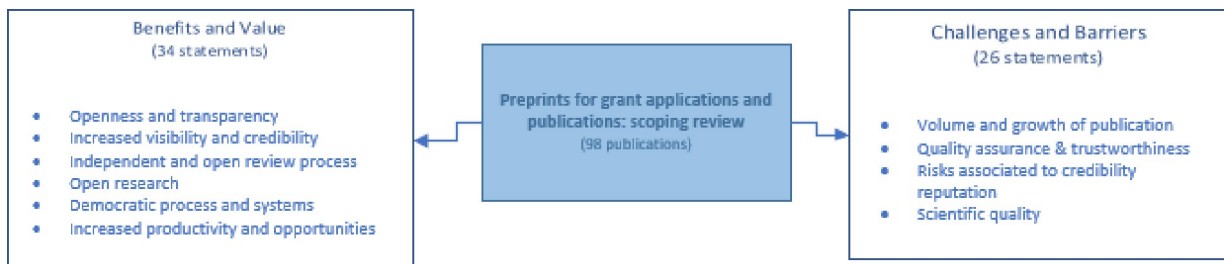

**Fig 2. Benefits and value statements grouped under six themes and challenges and barriers statements grouped under four themes.**

prioritised over quality or scrutiny [4, 46]. (See **Table 2** for a summary of statements associated to the barriers and challenges of preprints).

Without the proper guidance or caveats on preprint platforms, stating what they are and what they are not, there is the potential for preprints to have a negative impact on the credibility and public perception towards research and scientific discovery [2, 24, 27]. This is in part due to the public interpreting the preprint as 'established evidence' when it has not been through any peer review, scrutiny or quality assurance process [32]. These cautionary measures also extend to the use of preprints by clinicians and policymakers, especially if the study involves medications, vaccines or medical devices. The results could have a negative impact on the treatment of patients (including public health and safety) [7, 8, 26, 32, 33, 35, 45, 46, 48–51, 64] and result in the spreading of disinformation and distrust of research [24, 38].

## Benefits and value of preprints

A general consensus by those supportive of preprints was the notion of how preprints should be celebrated for bringing research faster to the public [2, 3, 8, 39, 69–73]. There was also acceptance that preprints are not a substitute for peer-reviewed publications, they are one element of the wider publication process of scientific discovery and sharing of research findings [3, 27, 74].

Most preprints do not differ substantially from the published peer-reviewed article, although there may be variation, there was no evidence that key sections of articles, such as results, changed between preprints and published versions [12, 24, 25, 36, 42, 75]. Brierley *et al.* (2021) found minimal changes were requested to preprint conclusions, suggesting that the entire publication pipeline is having a minimal but beneficial effect upon preprints and this was supported in the evidence [12, 24, 25, 36, 42, 75]. Moreover, the conclusions of 7.2% of non-COVID-19 related and 17.2% of COVID-19 related abstracts underwent only a discrete change (e.g., additional details on the funding statement) by the time of publication, but the majority of these changes do not qualitatively change the conclusions of the paper [10, 25].

As mentioned by Kaiser and others, intellectual 'scooping' and plagiarising have been regarded as a barrier to preprints [2, 12, 15, 47, 52–54]. However, this is unlikely to happen as preprints provide a time-sensitive/time-stamp when submitted to a preprint server [15, 47, 70]. As explained by Tennant *et al.* (2019), most preprints have a permanent identifier, often a Digital Object Identifier (DOI) which makes it easy to cite, track and establish a proof of discovery [47]. There is virtually no evidence to suggest research scooping exists from preprint servers [2, 41, 47, 76–78].

Key considerations found in the evidence suggested that preprints have the potential to offer free and immediate access to results, enabling the scientific community (including developed and emerging economies) to build upon research discovery [6, 26, 41, 50, 70, 78–81]. Preprints can facilitate feedback from a wider peer community than the traditional two to four reviewers needed to assess manuscripts submitted to journals [59]. This allows a unique platform for scientific critique and feedback before the work is subjected to more lengthy review processes conducted by a journal [61]. It also enables research to be openly accessible, reducing research waste and speeding up opportunities for future funding (using preprints as part of the grant application) [58, 69]. This not only has important implications for Early Career Researchers (ECRs) [53, 69] but also allows for negative findings to be published and openly accessible [5, 15, 50, 58, 59, 69, 70]. (See **Table 3** for a summary of statements associated to the benefits and value associated with the use of preprint publications).

The general consensus of those in support of preprints suggest that the focus has shifted to 'how preprints should be cited' rather than 'whether preprints should be cited' [35]. Furthermore, many of the potential risks associated with preprints reported not only from academia

but also from publishers and funding organisations, exist within the traditional peer review publication system [98]. The COVID-19 pandemic may have initiated a cultural shift in the use of preprints by academia, publishers, funding organisations, the public and policymakers as a whole [10, 25, 29, 30, 81].

## Academic community acceptability of preprints

One of the key challenges for the academic community is the lack of or no clear policies on the citation of preprints, from publishers and journals but also funding organisations. Klebel *et al.* (2020) found approximately 75% of journals do not have clear policies on co-reviewing, citation of preprints and publication of reviewers' identities for preprints [57]. As the evidence suggested there were several competing interests for academics, and these were more relevant in some disciplines than others and not all research needs to implement all open science practices due to the nature of the research [4, 59].

A survey conducted by Soderberg *et al.* (2020) found that of 3759 researchers (with 13.25% non-completion) 69.73% felt slightly to strongly favourable towards preprints, 15.16% felt opposed to preprints and 14.95% felt neutral, although there was variation between disciplines and academic career stages [9]. However, assessing the credibility of preprints is an important factor to researchers, such as the transparency of research content, accessibility to the data (information on the open research content) and an independent review and validation of the research claims [60].

For many investigators, preprints may be considered an initial step along the scientific dissemination and publication pathway, just as an abstract or poster have a role in the early sharing and dissemination of research among specialist communities before submission to a peer reviewed journal [26, 64]. However, preprints do not come without their challenges that can sometimes be in direct conflict with the pressures of research findings having to be accessed openly and transparently as early as possible [30]. These initiatives unintentionally place greater responsibility onto the scientific community and raise concerns that preprints are substandard work [12, 76]. However, there was no evidence to support such claims in the literature reviewed. Researchers will only publish their best work as it reflects their abilities and expertise, and often their reputation [70]. So, the notion of low quality would not be expected from those using preprints to disseminate their research findings [7, 23, 53, 59, 70]. Existing evidence from Brierley *et al.* (2021) and Watson (2022) for example have highlighted how preprints do not differ substantially from peer reviewed journal articles and since the COVID-19 pandemic, there has been a notable increase in the number of preprints [24, 25].

## Addressing the challenges of publication models

*"Despite the drawbacks and deadly consequences, there is little doubt that preprint publishing is here to stay. The question is how science will handle it. We are down a pathway of open science, and that pathway is going to accelerate. Our choice is not whether it occurs or not; our choice is whether it occurs responsibly." (Watson, 2022) [24].*

The evidence strongly indicated that the COVID-19 pandemic has changed the viewpoint and acceptance of preprints, at a time when rapid communication was needed and a greater visibility of effectively communicating the scientific advancements for healthcare [6, 8, 10, 99]. With more than half of the evidence being published during the pandemic suggests that in times of national emergency, initiatives such as open research and open access become even more under the spotlight [45]. In the first four months of the COVID-19 pandemic, 19,389 articles were available, with a third of these being preprints [24]. In fact, the media and government

quoted from preprints as the traditional peer review system was proving to be too slow [3, 27]. There was a surge of data insights during the pandemic that have ultimately accelerated the pace and speed of research, and having direct influence on policy and practice [10, 24]. However, as discussed by Raynaud *et al.* (2021) fast tracking peer review at a time of crisis can have implications to the quality of the assessment and overall results [32]. Rapid publication and potential risk of unverified (not being through a peer reviewed process) findings in health related research for example, can result in misinformation and misrepresentation of findings [15]. Teixeira da Silva noted several concerns around retractions and the threat preprints can impose on the validity of scientific discovery, especially at the height of the COVID-19 pandemic [34, 35]. Ethical guidance and disclaimers are needed to indicate where results from a preprint have not been peer reviewed or quality assured. Informing the reader of these caveats, may help members of the public to stay informed of the status of the evidence, which is particularly relevant for clinical trials and experimental research (**Box 1** provides an overview of the key considerations for best practice as summarised from the evidence).

### Box 1. Considerations for best practice

■ **Quality:** Preprints have quality controls including requirements for ethics approvals, consents, appropriate institutional archiving, and appropriate research reporting checklists

■ **Communications:** Multiple forms of communication are required at the right time, at the right stage and are translational to the wider public. Communicating in diverse ways and to diverse audiences requires implementing several straightforward ideas that includes the use of multimedia and digital media platforms

■ **Publishing:** Greater consistency and scrutiny across different journals or publishing platforms for medical scientific articles or manuscripts is needed to ensure quality and control are maintained even if different publishing models are used. Compatibility and appropriate checks on publication policies. Potential to reduce publication bias as not subject to publication selectivity.

■ **Open review:** Open assessment from a larger community enables diverse groups to contribute, offers a democratic process and opportunities to accelerate research discovery, priority, and dissemination

■ **Open access and inclusivity:** Open access of preprints immediately provides everyone with access to the research, including developing nations who often have limited funds (institutional or funding organisations) to publish, read, and subscribe to many publishing journals. Adopting preprints allows for enhancements to academic publishing and peer assessment to enhance scientific discovery and establish priority

■ **Diversifies Early Career Researchers:** Preprints can accelerate training, enable ECRs to publish through open access routes

■ **Ethics:** There is a need for principle-based regulations to achieve publication ethics for preprints. Gaining optimum accountability and transparency of research findings requires effort from both authors and preprint servers: (e.g., transparency in the time sequence of publication between preprints and subsequent peer-reviewed journal articles)

■ **Standards**: Stricter common standards for preprints are required that cover issues such as screening submissions and retracting those that turn out to be seriously flawed or fraudulent.

## Evidence from journals and publishers

There is growing acceptance from publishers recognising the value and necessity of preprints towards more open and transparent publication models [100]. Several articles included editorials or commentaries from journal publishers stating their position on preprints including changes to policy, and in the broader context of open access and open research [49, 71, 100–102]. Evidence from the articles included in the review have shown that preprints are becoming recognised as part of the dissemination publication process in the wider open research initiative, but to varying degrees based on clinical relevance [7, 32, 45].

Teixeira da Silva and Dobránszki (2019) conducted a review of preprint policies from 14 publishers and found that in 2017 64% of publishers had a preprint policy in place but by February 2018 this had increased to 78% [43]. Further advancements have been seen by eLife, showing that preprints can work alongside peer reviewed publications by adopting a 'publish, then review' model replacing the traditional 'review, then publish' model [3]. Approximately 70% of the papers reviewed by eLife during May, June and July 2020 had already been posted as preprints indicating that their model was already being implemented into practice *[45]*. **(See S3 Table: Publishers and Journals preprints policy or guidance).**

There is a clear indication that publishers are reviewing their policies and amplifying the preprints model [3, 45] and these changes were also evident in the grey literature of publishers and journals. Several publishers are actively encouraging the research community to use preprints to publish their manuscripts (e.g., Lancet, BMC, Thorax and eLife); offering to upload the manuscript to a preprint server on their behalf following submission (e.g., PLOS); developing and introducing their own preprint server for specific research (e.g., Nature for COVID research); reviewing tools to reduce information overload (e.g., simultaneous multi-preprint server retrieval tools to aggregate preprints); and publishing the preprint version alongside the peer reviewed journal article on the website (e.g., PeerJ and F1000Research) [4, 6, 23, 49, 74, 83, 98, 101].

Publishers such as the Lancet and eLife are also disseminating their own preprint policies and guidance in editorials and commentaries on how the model of publishing is changing [64]. The PubMed database is also taking steps to index preprints from journals such as F1000 and PeerJ. However, despite these developments, the evidence suggests that preprint policies vary between publishing companies and in some instances within the same publishing company and across research disciplines, which could hinder the progress and integrity of preprints [35, 40, 41, 42, 43, 57, 59]. **Box 2** provides a summary of the key considerations for publishers and journals as summarised from the evidence.

### Box 2. Summary of the key considerations for publishers

- Operating principles and mechanisms to enhance the publishing process and manuscript version control

- Enhanced cooperation between scholarly publishers and preprint servers

- Consider a peer review procedure for preprints

- When preprints are cited in peer-reviewed journals, they must be clearly indicated as such and remain under press embargo until published

- Formulate principles of publication ethics for conduct (e.g., release and use of preprints to serve as references

- A balance between immediacy and quality control is required to ensure research findings and new discoveries are not compromised due to process driven inefficiencies

- Opportunities to develop alternative publishing models whereby a two-step approach could ensure open access is truly promoted and credible.

### Evidence from funding organisations

Over the last five years several funding organisations have endorsed the use of preprints, encouraging researchers to use preprints as an interim to speed up the dissemination of the research [1]. There are also several funding organisations that allow the inclusion of preprints as part of a grant application, which are starting to be disseminated in published literature [70, 97]. However, the evidence was sparse, with most focusing on the publication aspect of research findings. **(See S4 Table: Funding organisations position on preprints for grant applications and publications).**

Some funding organisations have taken the initiative to develop their own exclusive platform to host preprints and other non-peer reviewed publications such as the Bill and Melinda Gates Foundation (platform to present findings as a future preprint server (Gates Open Research), using the F1000 Research technical platform that is leased for a fee [15]); Wellcome Trust (preprint server since 2016 (Wellcome Open Research)) [5]; and the National Institute for Health and Care Research (NIHR) (an Open Research platform for its researchers to publish, using prepublication checks and open peer review) [103].

The evidence also showed that many funding organisations are also assessing their policies and guidance or implementing new policies that take account of preprints in varying situations. A useful source to track funding organisations progress on these preprint policies is via the ASAPbio website [104, 105]. (see **Box 3 for** a summary of the key considerations for funding organisations based on the available evidence) [90].

---

### Box 3. Summary of the key considerations for funding organisations

- To develop coherent guidance for the performance evaluation of preprints

- To consider responsible use of metrics and deter from affiliation-based citation bias, including metrics of scholarly achievements

- Ensure that reviewers (both grant and publication reviewers) adhere to guidance on preprints, ensuring fairness and impartiality in the review process when evaluating performance of research

- Citation bias exists more with preprints; therefore, funding organisations need to deter from using these measures when making decisions about funding recommendations

- Implement policies to promote openness and ensure researchers are transparent about their procedures and share all aspects of their funded research (going beyond just publications), following the Declaration On Research Assessment (DORA)

- Regulating the use of preprints through the Plan U approach, mandating that funding organisations request that all grant applicants post their research first on preprints servers.

## Grant applications

There was limited evidence on the use of preprints in grant applications and most of the evidence was either in the grey literature or on the funding organisations webpages [15, 30, 70, 90].

Preprints can provide a funding organisation with evidence to substantiate recent productivity. However, there is wide variation in how funding organisations accept preprints, including in what circumstances they are acceptable in a grant proposal (e.g., researchers own preprints to demonstrate productivity or using preprints as part of the research proposal itself) [51, 70, 75]. Preprints are therefore said to have varying impact for researchers, grantees or applicants [2, 23, 41, 70, 75, 76, 78, 89, 90].

Several funding organisations (e.g., Wellcome Trust, NIH) now endorse the use of preprint servers for grant applications [106, 107]. Including preprints in the grant application process provides reviewers the opportunity to access and gain early insight to the research and data in its full format (i.e., not restricted by word limitation in the funding application) [3].

However, there have been concerns over the misuse of metrics, particularly with funding organisations using preprint citations [11, 60]. With the growing number of preprints there is a risk of author affiliation-based citation bias for those who use citations for scientific impact quantification (e.g., funding organisations, institutional promotion). This is particularly relevant to funding organisations who make recommendations to fund research based on citations. Although, we have seen a move away from this, with initiatives such as the Declaration on Research Assessment (DORA) [108], Contributor Roles Taxonomy (CRediT) [109] and Plan S/cOAlition S [110] this is not a global phenomenon [30, 60] and has important implications for diversity, equity and inclusion (including ECRs and Low-and-Middle Income Countries (LMICs)) [23, 30, 53, 59, 60].

## Discussion

It was clear from the evidence that there was a general acceptance and understanding that preprints are not a substitution or replacement for a peer reviewed publication. Preprints complement and act as a mechanism within the publication process to disseminate the research findings faster, through openly accessible routes [23, 55, 92].

The distinction between a preprint and a peer reviewed publication becomes imperative when you consider the unintended consequences of their use on not only clinical practice but also on researchers to judge research performance, quality or productivity [2, 29, 64, 70, 91]. Whilst preprints can be part of the process to promote open research, by applying the principles of transparency and integrity, we also need to appreciate and acknowledge the associated caveats and limitations of their use [25, 26, 33, 64].

Although the COVID-19 pandemic has informed a greater appreciation for the scholarly landscape on the use and acceptability of preprints, there have been withdrawals of several preprints [10, 34]. However, there is no way of knowing whether peer review would have identified the problems resulting in the withdrawal of the manuscript at the point of publication. This is also true for peer reviewed journal articles, as articles are only withdrawn or retracted once published (issues raised by readers or the authors themselves). These challenges are not exclusive to preprints, however with careful consideration and acceptance of these challenges, alongside the limitations, preprints can become an integral part of scientific discovery if all journals support publishing on preprint servers as part of their process (not all journals ask authors to submit their manuscript on a preprint server as part of the submission process to a peer reviewed journal) [6]. Evidence comparing preprints to peer reviewed journal articles

have found that the quality is similar for both, and the types of discrepancies (e.g., study characteristics, outcome reporting, numerical and statistical values) are common between both preprints and a peer reviewed journal article [10, 99, 111–114]. Opportunities for rapid dissemination using preprints means that best practices through policies and guidance are required, from journals, publishers, and funding organisations, to ensure that preprints become embedded into practice albeit with caveats and cautionary measures to maintain quality, credibility, transparency and openness (as with peer reviewed articles submitted to journals). Initiatives such as Plan S (open access publishing) and cOAlition S (funders and stakeholders) advocate for immediate and open access to research publications without embargo [110], which over time may ultimately reassure the research community of the value and benefit of publishing on a preprint server prior to the submission of the manuscript to a journal.

The increasing demands and challenges to allow for the immediacy of results being made public places pressure on the open access agenda for not only publishers but also the research community [87]. Delays to the publication of manuscripts can also have far reaching implications for ECRs in terms of job opportunities and grant funding [53, 60, 61, 80]. Understanding the complexities that surround the use, acceptability, challenges and benefits of preprints goes far beyond the world of publishing research: policies and practices of funding organisations and university institutions have a direct impact on the longer-term impact of how research enhancement and discovery takes place [7, 23, 51, 59, 70].

## Study strengths and limitations

The main strength of the review was the combination of a systematic database literature search with an extensive grey literature search of publishers and funding organisations position, policies, and guidance. However, as scoping reviews map the evidence rather than assess quality or risk of bias, this was a limitation to the current review. Restricting the evidence to only a health and social care discipline meant that articles from other disciplines were not included in the review, with the majority of evidence focused on publishing models and minimal evidence looking at the role of preprints in grant applications. Although, there were some articles from Asia (12.3%), most of the articles were from Europe and Americas (72.4%), which could have introduced regional biases (e.g., Western science). However, this could be due to a lack of evidence from these regions rather than missing literature from the systematic searches. Only searching in two databases could have meant we missed some articles (including preprints) on this subject matter. However, the role of scoping reviews are an overview of a broad topic or subject (identifying key concepts, gaps in the evidence and sources of evidence), rather than to evaluate or synthesise data on a focused research question or topic [18].

Due to the limited evidence found in the review on the acceptability and use of preprints in grant applications, research is needed to clearly understand what position funding organisations are taking and how it is perceived and accepted by the research community (with initiatives such as Plan S and cOAlition S [110]). There was a lack of evidence to clearly explain in what circumstances preprints are acceptable to cite in the grant application process. For example, to inform the research application or as part of an applicant's CV, at what stage is it acceptable to include and what version of record, and how is it tracked when published or otherwise as a peer reviewed journal article (particularly for funding organisations).

## Future recommendations

The recommendations arising from the review mainly focused on what measures are needed for preprints to become embedded in open research practices. These recommendations are

important for publishers, funding organisations and researchers, by considering the role of preprints and where and in what circumstances preprints may not be feasible, such as

- Accessibility and feasibility to submit and upload to preprint servers on behalf of the researcher (therefore ensuring continuity) [78]

- Adopting the use of preprints in grant applications requires clear and ethical guidance

- Applying a badge or signal to show whether the article complying with open data, data sharing, pre-registrations etc could also be adopted by preprint services (with clear guidance) similar to the Center for Open Science (COS) (a badge system has led to over 60 journals adopting this approach on published articles to indicate open data, data sharing, materials, and/or pre-registrations) [9, 33]

- Preprint servers must have clear ethical and retraction policies in place, and these must be enforced and considered in publishers and funding organisations preprint policies [15, 34, 115]

- Preprint policies should include ethical guidance in the event of misconduct and consider whether this should be the same process as peer-reviewed journal articles [34]

- Preprints should not be posted when the reported information could be misapplied or misused, causing significant consequences to the health and safety of the public [116]

(See S5 Table: Future considerations and recommendations for publishers, funding organisations and the research community).

Clear, explicit, and transparent guidance by publishers, journals, and funding organisations (working collaboratively) are required to articulate to academia the role and purpose of preprints compared to a peer reviewed journal publication. There is also a need for publishers and journals to acknowledge how preprints fit into their publication model, ensuring that policies are aligned with preprints servers and are transparent about their use of preprints.

Methodologically, it would be important to consider preprints in other disciplines other than health and social care, to understand whether there are differences between disciplines and whether those that submit and have used preprint servers for several years, approach preprints with the same level of caution or not.

## Conclusions

Further research is needed to address several key areas associated with preprints that may help to clarify the respective opportunities that preprints can offer to the publication process. It will be important to consider the views, opinions and expectations of researchers, funding organisations and publishers, to ensure that global challenges regarding access to research findings (e.g., budgetary constraints) does not inhibit ECRs or LMICs [23, 53, 84]. The lack of evidence on preprints at the grant application stage requires more research with funding organisations and the research community, such as screening of preprints submissions used in grant applications as part of the evidence to support the need of the research and as part of the applicants' publications listing [15, 43, 70].

The question of quality, peer review, and integrity are paramount to the acceptability of preprints if they are to contribute to improving the reproducibility of research and further advance scientific discovery [50]. It is clear from the evidence that a clearer understanding of the role of preprints in the process of publishing research findings is needed [55]. Possible coordination among preprint servers for screening research and opportunities to work with funding organisations and researchers to increase their longer-term functionality [23, 66].

There is acceptance that preprints offer a way to present early findings of research into the public domain, openly and transparently, which can therefore help to foster awareness and reduce misinformation of health and social care research findings to the general public [112, 117]. The role of preprints is merely part of the process to reaching a peer reviewed publication. Preprints are therefore not a substitute for peer reviewed journal publications (and not all peer reviewed publications are published in a journal, and peer review can happen outside the context of journals such as Peer Community in [118]) but form part of the overall system and process of disseminating research rapidly using openly accessible routes to advance scientific discovery [12, 23].

Any changes to the processes and mechanisms of publishing research needs to ensure it meets current expectations of different audiences, and that research communities embrace the principles of quality, transparency and openness for the provision of innovation and new knowledge [31].

## Supporting information

**S1 Appendix. Search terms and keywords.**
(DOCX)

**S2 Appendix. PRISMA checklist.**
(DOCX)

**S1 Table. Scopus and Web of Science searches.**
(DOCX)

**S2 Table. Full details of the 98 included studies.**
(DOCX)

**S3 Table. Publishers and Journals preprints policy or guidance.**
(DOCX)

**S4 Table. Funding organisations position on preprints for grant applications and publications.**
(DOCX)

**S5 Table. Future considerations and recommendations for publishers, funding organisations and the research community.**
(DOCX)

## Acknowledgments

We would like to thank the NIHR Journals Library centre staff and those members of the NIHR Open Research Oversight Group for their support during the development of the scoping review and throughout the write up of the article. We would also like to thank the NIHR Research on Research team for reviewing early draft versions of the article.

## Author Contributions

**Conceptualization:** Amanda Jane Blatch-Jones, Alejandra Recio Saucedo.

**Data curation:** Amanda Jane Blatch-Jones, Alejandra Recio Saucedo, Beth Giddins.

**Formal analysis:** Amanda Jane Blatch-Jones, Alejandra Recio Saucedo, Beth Giddins.

**Investigation:** Amanda Jane Blatch-Jones, Alejandra Recio Saucedo, Beth Giddins.

**Methodology:** Amanda Jane Blatch-Jones, Alejandra Recio Saucedo.

**Project administration:** Amanda Jane Blatch-Jones.

**Supervision:** Amanda Jane Blatch-Jones.

**Writing – original draft:** Amanda Jane Blatch-Jones.

**Writing – review & editing:** Amanda Jane Blatch-Jones, Alejandra Recio Saucedo, Beth Giddins.

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
