## [Decision Letter · Decision Letter 0]

17 May 2023

PONE-D-23-09198A scoping review on the use and acceptability of preprintsPLOS ONE

Dear Dr. Blatch-Jones,

Thank you for submitting your manuscript to PLOS ONE. After careful consideration, we feel that it has merit but does not fully meet PLOS ONE’s publication criteria as it currently stands. Therefore, we invite you to submit a revised version of the manuscript that addresses the points raised during the review process.

We look forward to receiving your revised manuscript.

Kind regards,

Robin Haunschild

Academic Editor

PLOS ONE

Journal Requirements:

   "This research was funded by the National Institute for Health and Care Research Coordinating Centre (NIHRCC), based at the University of Southampton, through its Research-on-Research Programme. The views and opinions expressed in the discussion are those of the authors and do not necessarily reflect those of the Department of Health and Social Care."  

Additional Editor Comments:

As it is the PLOS policy of not accepting (subjective) reviews but systematic reviews, it might be better to resubmit this manuscript in a revised form as a systematic review. I do not see that the manuscript includes enough original research for a research article. However, you may feel free to provide your reasons for resubmitting as a research article in a cover letter if you prefer to do so. In any case, please carefully consider the comments of the reviewers and provide point-by-point responses in a rebuttal letter.

Reviewers' comments:

Reviewer's Responses to Questions

**Comments to the Author**

1. Is the manuscript technically sound, and do the data support the conclusions?

Reviewer #1: Yes

Reviewer #2: Yes

2. Has the statistical analysis been performed appropriately and rigorously? 

Reviewer #1: N/A

Reviewer #2: N/A

3. Have the authors made all data underlying the findings in their manuscript fully available?

Reviewer #1: Yes

Reviewer #2: Yes

4. Is the manuscript presented in an intelligible fashion and written in standard English?

Reviewer #1: Yes

Reviewer #2: Yes

5. Review Comments to the Author

Reviewer #1: Batch-Jones et al have undertaken a scoping review into the use and acceptability of preprints in the biosciences. The authors review 98 articles in total and provide a good summary of the opinions relating to how scientists view preprints, both positively and negatively. Overall this is an important review that provides tangible suggestions and considerations for different stakeholders and is a highly beneficial addition to the field.

I do not have any major concerns however I do believe the following would improve the manuscript:

The biggest element that I feel is missing from this review is the evidence that exists comparing preprints to their peer reviewed counterparts. This addition would particularly strengthen the argument against the notion that “preprints are substandard work” (lines 342-346). This is also relevant to the point made in Table 2 relating to “quality assurances and trustworthiness”. Further commenting on the various preprint peer-review platforms would be a nice addition to reflect where the field is moving – over 6000 preprints have so far been reviewed in this manner and this is something that funders are increasingly considering. For a starting list of papers I think are missing or would add to this discussion;

• Bero et al

• Carneiro et al

• Fleerackers et al

• Klein et al

• Pagliaro et al

• Sumner et al

• Nicholson et al

These papers may have been excluded by the authors during their selection criteria but I do feel they would add to the impact of this review.

Limitations – a discussion on the limits with regards to the highly biased “Western science” literature. I note that the authors only included a single study focussed on the African continent. This is not as a result of the authors missing a large amount of literature – the literature simply is lacking for this region – but it’s a very important limitation and point of discussion. One of the major benefits of preprints is that they enable researchers to share and read work without paying APCs or access charges that they aren’t able to afford, thereby increasing equity in the publishing process.

The authors make numerous mentions to the importance and role of preprints during the COVID19 pandemic, referring to the pandemic as initiating a "cultural shift" in the use of preprints, which is direct reference to our work which has not been cited (Fraser et al, PLOS Biology) whereas a nature news piece (which took heavily from out work) has been cited in other places to support this statement. I think these statements should be supported with references to the literature and I'm happy for the authors to cite work other than my own as there are other studies that support this should they so choose.

Reviewer #2: My review is available at see https://ludowaltman.pubpub.org/pub/review-scoping-review-preprints/release/2.

As a signatory of Publish Your Reviews, I have committed to publish my peer reviews alongside the preprint version of an article. For more information, see http://publishyourreviews.org.

6. PLOS authors have the option to publish the peer review history of their article (what does this mean?). If published, this will include your full peer review and any attached files.

Reviewer #1: **Yes: **Jonathon A Coates

Reviewer #2: **Yes: **Ludo Waltman

---

## [Author Response · Author response to Decision Letter 0]

20 Jul 2023

All of my responses to the editor and peer reviewer comments are addressed in a separate document included in the attached files due to the word count and table format.

---

## [Decision Letter · Decision Letter 1]

17 Aug 2023

PONE-D-23-09198R1The use and acceptability of preprints in health and social care settings: a scoping reviewPLOS ONE

Dear Dr. Blatch-Jones,

Thank you for submitting your manuscript to PLOS ONE. After careful consideration, we feel that it has merit but does not fully meet PLOS ONE’s publication criteria as it currently stands. Therefore, we invite you to submit a revised version of the manuscript that addresses the points raised during the review process.

We look forward to receiving your revised manuscript.

Kind regards,

Robin Haunschild

Academic Editor

PLOS ONE

Journal Requirements:

Additional Editor Comments:

Please revise your manuscript according to the suggestions by reviewer #2.

Reviewers' comments:

Reviewer's Responses to Questions

**Comments to the Author**

1. If the authors have adequately addressed your comments raised in a previous round of review and you feel that this manuscript is now acceptable for publication, you may indicate that here to bypass the “Comments to the Author” section, enter your conflict of interest statement in the “Confidential to Editor” section, and submit your "Accept" recommendation.

Reviewer #1: All comments have been addressed

Reviewer #2: (No Response)

2. Is the manuscript technically sound, and do the data support the conclusions?

Reviewer #1: Yes

Reviewer #2: Yes

3. Has the statistical analysis been performed appropriately and rigorously? 

Reviewer #1: Yes

Reviewer #2: N/A

4. Have the authors made all data underlying the findings in their manuscript fully available?

Reviewer #1: Yes

Reviewer #2: Yes

5. Is the manuscript presented in an intelligible fashion and written in standard English?

Reviewer #1: Yes

Reviewer #2: Yes

6. Review Comments to the Author

Reviewer #1: The authors ahve addressed my comments and I'd recommend the manuscript for publication. This is an important addition to the field.

Reviewer #2: Please see my review report available online at https://ludowaltman.pubpub.org/pub/review-scoping-review-preprints2/release/1.

7. PLOS authors have the option to publish the peer review history of their article (what does this mean?). If published, this will include your full peer review and any attached files.

Reviewer #1: **Yes: **Jonathon Alexis Coates

Reviewer #2: **Yes: **Ludo Waltman

---

## [Author Response · Author response to Decision Letter 1]

31 Aug 2023

Responses to the decision letter and reviewer comments are provided in the submitted response to reviewers document

---

## [Decision Letter · Decision Letter 2]

4 Sep 2023

The use and acceptability of preprints in health and social care settings: a scoping review

PONE-D-23-09198R2

Dear Dr. Blatch-Jones,

We’re pleased to inform you that your manuscript has been judged scientifically suitable for publication and will be formally accepted for publication once it meets all outstanding technical requirements.

Kind regards,

Robin Haunschild

Academic Editor

PLOS ONE

Additional Editor Comments (optional):

Reviewers' comments:

Reviewer's Responses to Questions

**Comments to the Author**

1. If the authors have adequately addressed your comments raised in a previous round of review and you feel that this manuscript is now acceptable for publication, you may indicate that here to bypass the “Comments to the Author” section, enter your conflict of interest statement in the “Confidential to Editor” section, and submit your "Accept" recommendation.

Reviewer #2: All comments have been addressed

2. Is the manuscript technically sound, and do the data support the conclusions?

Reviewer #2: Yes

3. Has the statistical analysis been performed appropriately and rigorously? 

Reviewer #2: N/A

4. Have the authors made all data underlying the findings in their manuscript fully available?

Reviewer #2: Yes

5. Is the manuscript presented in an intelligible fashion and written in standard English?

Reviewer #2: Yes

6. Review Comments to the Author

Reviewer #2: (No Response)

7. PLOS authors have the option to publish the peer review history of their article (what does this mean?). If published, this will include your full peer review and any attached files.

Reviewer #2: **Yes: **Ludo Waltman

---

## [Editor Report · Acceptance letter]

8 Sep 2023

PONE-D-23-09198R2 

The use and acceptability of preprints in health and social care settings: a scoping review 

Dear Dr. Blatch-Jones:

I'm pleased to inform you that your manuscript has been deemed suitable for publication in PLOS ONE. Congratulations! Your manuscript is now with our production department. 

Kind regards, 

on behalf of

Dr. Robin Haunschild 

Academic Editor

PLOS ONE